# Flavor Characteristics of Ten Peanut Varieties from China

**DOI:** 10.3390/foods12244380

**Published:** 2023-12-05

**Authors:** Bin Ding, Fei Wang, Bei Zhang, Mengshi Feng, Lei Chang, Yuyang Shao, Yan Sun, Ying Jiang, Rui Wang, Libin Wang, Jixian Xie, Chunlu Qian

**Affiliations:** 1Taizhou Institute of Agricultural Sciences, Jiangsu Academy of Agricultural Sciences, Taizhou 210014, China; 20172201@jaas.ac.cn (B.D.); 20172202@jaas.ac.cn (M.F.); 20142207@jaas.ac.cn (L.C.); 20142204@jaas.ac.cn (Y.J.); wangruishiwei@163.com (R.W.); 2Department of Food Science and Engineering, School of Food Science and Engineering, Yangzhou University, Yangzhou 225012, China; wfei1101@163.com (F.W.); zhbeiang06@163.com (B.Z.); qq1046769248@163.com (Y.S.); 18705276730@163.com (Y.S.); 3College of Food Science and Technology, Nanjing Agricultural University, Nanjing 210095, China; wanglibin@njau.edu.cn

**Keywords:** peanut, flavor, free amino acids, 5′-nucleotides, volatiles

## Abstract

To investigate the flavor characteristics of peanuts grown in Jiangsu, China, ten local varieties were selected. The amino acids, 5′-nucleotides and volatile substances were detected, and the flavor and odor characteristics of these varieties were estimated using an electronic tongue and nose. The results showed that the fat and protein contents of ten peanut varieties changed significantly (*p* < 0.05), and may have been negatively correlated with those of the Taihua 6 variety—in particular, having the highest protein content and the lowest fat content. The amino acid contents of the peanuts were 20.08 g/100 g (Taihua 4)–27.18 g/100 g (Taihua 6). Taihua 6 also contained the highest bitter (10.41 g/100 g) and sweet (6.06 g/100 g) amino acids, and Taihua 10 had the highest monosodium glutamate-like amino acids (7.61 g/100 g). The content of 5′-nucleotides ranged from 0.08 mg/g (Taihua 9725) to 0.14 mg/g (Taihua 0122–601). Additionally, 5′-cytidylate monophosphate (5′-CMP) and 5′-adenosine monophosphate (5′-AMP) were the major 5′-nucleotides detected in the peanuts. A total of 42 kinds of volatile flavor compounds were detected, with both Taihua 4 and 6 showing the most (18 kinds) and the highest content being in Taihua 4 (7.46%). Both Taihua 9725 and 9922 exhibited the fewest kinds (nine kinds) of volatile components, and the lowest content was in Taihua 9725 (3.15%). Formic acid hexyl ester was the most abundant volatile substance in peanuts, and the highest level (3.63%) was detected in Taihua 7506. The electronic tongue and nose indicated that the greatest taste difference among the ten varieties of peanuts was mainly related to sourness, and Taihua 4 and Taihua 9922 had special taste characteristics. On the other hand, the greatest smell difference among the ten varieties of peanuts was mostly for methane and sulfur organic substances, and Taihua 0605-2 had a special and strong smell characteristic. In conclusion, the content and composition differences of the flavor substances of ten peanut varieties were responsible for their divergences in taste and smell. These results will provide guidelines for the further use (freshly consumed or processed) of these ten peanut varieties.

## 1. Introduction

Peanuts (*Arachis hypogaea* L.) are among the most important economic crops. The seeds of peanuts can be eaten directly or used for oil, to make butter or in a variety of dishes [1]. China is the main producer and processor of peanuts, accounting for more than half of global peanut production; hence, peanuts are the main crop in many places across China. Because of the high consumption, China imports more peanuts than it exports, so the self-cultivation of peanuts is important in China. As the yield of peanuts increases, the evaluation of peanuts begins to shift from quantity to quality. The main ingredients of peanuts are fats and proteins, along with health-beneficial substances (oleic acid, vitamin E, folic acid, niacin, flavonoids, isoflavones) [2,3], and these components influence the flavor characteristics of peanuts. As a type of pulse food, peanuts can be freshly consumed and the flavor of peanuts directly influences their acceptance by consumers. As a raw material for other food production, peanuts can be consumed after processing or cooking, and the flavor of peanuts affects the flavor characteristics of processed or cooked foods, as well as their acceptance [4,5]. Therefore, the flavor of peanuts is one of the most vital commercial characteristics considered during the breeding, cultivation and processing of these pulses. The flavor of raw plant food materials, including peanuts, can be greatly affected by factors such as variety [6], location [7], harvest season and ripening stage [8], among which the variety has the greatest impact on the flavor characteristics of plant products, as the variety impacts the differences in genetic material.

Peanuts comprise over 500 varieties, each of which exhibits different planting characteristics and quality representations [9]. The flavor characteristics of certain peanut varieties are unique and heritable [10,11,12], so a comparative analysis of the flavors of similar varieties is necessary for planting and processing. Peanut varieties are discreetly and carefully selected for butter processing purposes based on their content and composition regarding color, flavor, nutritional substances and market acceptance [5,13]. Their flavor components comprise non-volatile and volatile substances, free amino acids and 5′-nucleotides as the main non-volatile flavor materials. These determine the taste characteristics of foods, and small-molecule volatile substances dominate the characteristic smells of foods. The flavor characteristics of foods are usually estimated via sensory evaluation, but high subjectivity and regional characteristics mean that the evaluation is not objective. The electronic tongue and nose equipped sensors can detect flavor substances and produce response values that express flavor characteristics in a quantitative manner, making this method much more objective and credible for evaluating flavor [14]. High-performance liquid chromatography (HPLC) and headspace solid-phase micro-extraction gas chromatography (HS–SPME–GC–MS) are efficient and well-established methods for separating various non-volatile and volatile flavor substances, and were previously successfully applied in the detection of flavor substances in coffee [15], meat [16] and lotus roots [14].

In this study, the flavors of ten Chinese peanut varieties were compared, their non-volatile and volatile flavor components were identified and their flavor characteristics were estimated using an electronic tongue and nose. The obtained results established the variety-specific flavor profiles of the peanuts from Jiangsu, China, and also revealed the flavor differences between ten varieties of peanuts, forming the basis for the further research and utilization (fresh, processed or cooked) of these ten peanut varieties.

## 2. Materials and Methods

### 2.1. Raw Materials

Ten varieties of fresh peanuts (*Arachis hypogaea* L.) were collected from the pilot farm of Taizhou Institute of Agricultural Science, Jiangsu Academy of Agricultural Sciences, Jiangsu, China. The ten varieties of peanuts were named ‘Taihua0122-601’, ‘Taihua6’, ‘Taihua0311’, ‘Taihua0605-2’, ‘Taihua10’, ‘Taihua11’, ‘Taihua7506’, ‘Taihua9725’, ‘Taihua4’ and ‘Taihua9922’. All the varieties of peanuts are widely cultivated in China, and were collected and planted for a deep comparative analysis (production, resistance, flavor, processing, etc.) on the same pilot farm. All the fresh peanuts were dried under natural conditions (30–35 °C) and peeled, and the seeds were collected separately. Then, they were sealed in plastic bags (polyethylene, oxygen permeability, 12.5 cm^3^ m^−2^ h^−1^ mpa^−1^) and stored at 4 °C with 50% relative humidity for a further analysis 1 week later. All the measurements were replicated in triplicate.

### 2.2. Physicochemical Properties of Peanuts

The water content of the peanuts was measured via further dehydration in a drying box (Yiheng DHG-9070A, Shanghai Yiheng Medical Equipment Co., Ltd., Shanghai, China) at 105 °C until a constant weight was obtained, and the difference between the weight before and weight after drying was recorded as the water content. The fat content of the peanuts was measured using the Soxhlet extraction method [17]. After the measurement of the water content, the dried sample (5 g) was crushed into powder (200 mesh pellets), wrapped in a filter paper cartridge and placed in a Soxhlet extractor for the total extraction of the fat. The protein content of the peanuts was measured using an automatic Kjeldahl nitrogen analyzer (Haineng K9840, Jinan Haineng Instrument Co., Ltd., Jinan, China). All operations followed the manufacturers’ instructions for equipment use.

### 2.3. Determination of Non-Volatile and Volatile Components

The non-volatile (free amino acids and 5′-nucleotides) and volatile components of the peanuts were measured via HPLC and HS–SPME–GC–MS, and all the procedures were carried out employing the method that we previously developed [14]. A total of 1 g of the peanut sample was ground with distilled water, and incubated at 80 °C for 20 min (amino acids) or 100 °C for 5 min (5′-nucleotides). Then, the sample was topped up to a 10 mL volume after cooling to room temperature, and filtered using a 0.45 μm nylon membrane. During the extraction, both amino acids and 5′-nucleotides were separated on an inertsil–ODS–SP–C18 column (250 × 4.6 mm, Shimadzu, Kyoto, Japan), and determined via Waters E2695 series HPLC (Waters Technologies, Milford, MA, USA). A total of 3 g of the peanut sample was ground with 3 mL of sodium chloride (saturation) and 10 μL of 1pyr2-dichlorobenzene as the internal standard and put into a 20 mL glass bottle. The bottle was sealed, and the contents were mixed via magnetic stirring. Polydimethylsiloxane (PDMS) (65 μm) (preheated at 250 °C for 20 min) was inserted into the headspace (HS) at 55 °C for 30 min and then into the injection port of the GC–MS. The volatiles were separated and determined by employing the GC–MS apparatus (Trace TSQ, Themo Fisher Scientific, Waltham, MA, USA). The detected peaks and areas were compared against the National Institute of Standards and Technology (NIST) Standard Reference Database 78, and the calculations were performed using the peak area normalization method.

### 2.4. Determination of Taste and Smell Characteristics via Electronic Tongue and Nose

The taste and smell characteristics of the peanuts were estimated using the electronic tongue and nose, and all the procedures were carried out using the method we previously developed [14]. A total of 5 g of the peanut sample was ground with distilled water, topped up to a 50 mL volume, incubated at 55 °C and stirred for 10 min. After centrifugation at 3000× *g* for 10 min, the supernatant was collected and filtered, and then added to the electronic tongue (Shanghai Bosin Industrial Development Co., Ltd., Shanghai, China) to analyze the taste characteristics of the peanuts. A total of 5 g of the peanut sample was ground with distilled water (10 mL), and the homogenate was incubated in a sample bottle at 55 °C for 20 min. The headspace air was collected and analyzed using the electronic nose (Shanghai Bosin Industrial Development Co., Ltd.).

### 2.5. Statistical Analyses

All measurements were replicated in triplicate, and the data are expressed as the mean ± standard deviation (mean ± SD). A principal component analysis (PCA) was performed using RStudio (2023.07.0), a radar fingerprint analysis was performed using Origin and a variance (ANOVA) analysis was performed using SPSS Statistics 26.

## 3. Results

### 3.1. Physicochemical Properties of Ten Peanut Varieties 

The water content of the peanuts was 4.752–5.483%, with the lowest being in Taihua 0605-2 and the highest in Taihua 11. The fat content of the peanuts was 48.24–53.83%, with the lowest in Taihua 6 and the highest in Taihua 0605-2. The protein content of the peanuts was 23.95–29.90%, the lowest being in Taihua 7506 and the highest in Taihua 6 (Table 1).

### 3.2. Free Amino Acid Contents of Ten Peanut Varieties

As shown in Table 2, the total amino acid contents of the peanuts of ten varieties ranged from 20.08 g/100 g (Taihua 4) to 27.18 g/100 g (Taihua 6). The single amino acid contents were 0.39 g/100 g (L-tyrosine) to 3.50 g/100 g (L-aspartic). The bitter amino acid (histidine, arginine, valine, methionine, phenylalanine, isoleucine and leucine) contents were 5.90 g/100 g (Taihua 4) to 10.41 g/100 g (Taihua 6), accounting for 29.38% and 38.30% of the total amino acid contents in the peanuts, respectively. The sweet amino acid (glycine, serine, threonine and alanine) contents were 3.47 g/100 g (Taihua 0311) to 6.06 g/100 g (Taihua 6), accounting for 16.12% and 22.30% of the total amino acid contents in the peanuts, respectively. The monosodium glutamate (MSG)-like amino acid (aspartic, glutamic) contents were 4.77 g/100 g (Taihua 0605-2) to 7.61 g/100 g (Taihua 10), accounting for 20.99% and 28.78% of the total amino acid contents in the peanuts, respectively.

### 3.3. 5′-Nucleotide Contents of Ten Peanut Varieties

As shown in Table 3, the 5′-nucleotide contents ranged from 0.08 mg/g (Taihua6, 10, 9725) to 0.14 mg/g (Taihua0122-601). The compounds 5′-cytidylate monophosphate (5′-CMP) and 5‘-adenosine monophosphate (5′-AMP) comprised the main 5′-nucleotides in the peanuts, with Taihua7506 being the only variety with no 5′-AMP detected. On the other hand, 5′-uridine monophosphate (5′-UMP) was only detected in Taihua0311 and Taihua10; 5′-guanosine monophosphate (5′-GMP) was only detected in Taihua0122-601, Taihua0605-2, Taihua11, Taihua7506 and Taihua9922; and 5′-inosine monophosphate (5′-IMP) was not detected in any of the peanut varieties.

### 3.4. Volatile Aroma Components of Ten Peanut Varieties 

As shown in Table 4, a total of 42 kinds of volatile flavor compounds were identified within the ten peanut varieties, with 14, 18, 14, 15, 11, 9, 10, 9, 18 and 9 kinds of volatile compounds detected in Taihua0122-601, 6, 0311, 0605-2, 10, 11, 7506, 9725, 4 and 9922, respectively, and the total contents were 3.23, 4.94, 3.37, 6.39, 5.37, 6.54, 5.80, 3.15, 7.46 and 3.47%, respectively. Formic acid hexyl ester was the most abundant volatile substance in peanuts, and the highest content (3.63%) was identified in Taihua 7506. Only 1-nonanol, nonanal and 3-methyl-undecane were detected within all ten varieties of peanuts.

### 3.5. Electronic Tongue for the Taste Characteristics of Peanuts

As shown in Figure 1, the sourness, bitterness, astringency, umami and saltiness of the ten peanut varieties were estimated using the electronic tongue, and all the peanuts showed high umami, saltiness, astringency and bitterness response values, and low sourness response values. The greatest taste difference between the ten peanut varieties was found for the response values of sourness, whereby Taihua 0311 showed the lowest value and Taihua 4 the highest. There were few differences among the response values for bitterness, astringency and umami. The flavor characteristics of the ten peanut varieties were effectively separated via the principal component analysis (PCA); the contribution rate of PCA1 was 97.06%, and the contribution rate of PCA2 was 2.63%, totaling 99.69% (Figure 2), which indicated that the two-dimensional scatter diagram of the PCA could reflect the taste differences between the ten varieties of peanuts. There were overlaps between ten varieties of peanuts, indicating great similarity regarding the tastes of the ten varieties, but there were still differences between Taihua 4, Taihua 9922 and other varieties. The loading plot of the PCA also indicated that the taste contributions of the peanuts were in the order of sourness, astringency, bitterness, umami and saltiness (Figure 3), and the group differences were mostly related to sourness (Figure 2 and Figure 3).

### 3.6. Electronic Nose for the Smell Characteristics of Peanuts

As shown in Figure 4, the smell characteristics of the ten peanut varieties were estimated using an electronic nose equipped with 10 kinds of sensors (Table 5), and the W1S, W1W, W2W and W5S sensors showed high response values, with Taihua0605-2 exhibiting the highest values. As shown in Figure 5, the contribution rate of PCA1 was 99.7% and the contribution rate of PCA2 was 0.01%, totaling 99.71%, which indicated that the two-dimensional scatter diagram of the PCA could effectively reflect the odor differences between ten varieties of peanuts. No overlaps appeared within PCA1 and PCA2, indicating that the smell characteristics differed across the ten varieties of peanuts, whereby Taihua 0605-2 exhibited significantly different smell characteristics compared to the other varieties, with a much smaller PCA1 value compared to that of the other varieties. The loading plot indicated that the smell contributions for the peanuts were in the order of W1S, W2S, W5S, W2W, W3S, W6S, W1W, W5C, W1C and W3C (Figure 6), and the reference substances were in the order of methane, carbon monoxide, nitrogen dioxide, hydrogen sulfide, methane, hydrogen, hydrogen sulfide, propane, toluene and benzene (Table 5).

## 4. Discussion

The water contents of the ten peanut varieties showed no significant (*p* > 0.05) differences. Taihua 0605-2 was the only variety exhibiting a significantly (*p* < 0.05) lower water content than Taihua 11 (Table 1), indicating that the degree of dehydration in peanuts before storage was similar. Moreover, Taihua 0605-2 peanuts may not require such stringent drying conditions (thin shell; small fruit) compared to the other varieties. The fat and protein contents were negatively correlated, especially for Taihua 6, which exhibited the highest protein content and lowest fat content. This is directly affected by hereditary properties. The protein and fat contents in this study are similar to those previously reported [18,19]. The non-volatile aroma compounds were mainly proteins and their degradation products (amino acids or polypeptides), and the volatile aroma compounds were fats (small-molecule lipid substances such as formic acid hexyl ester). Thus, the differences in the content and composition of proteins and fats in peanuts, largely influenced by the genetic material, are of crucial importance for their flavor [20].

Amino acids were the main flavor substances, manifesting different taste characteristics (bitter, sweet and MSG-like tastes) [21]. They also comprised the basic protein components [22]; hence, a higher protein content in peanuts also implied a higher amino acid content. Taihua6 peanuts showed the highest protein, amino acid and bitter and sweet amino acid contents. The differences in amino acid contents between the ten peanut varieties were affected by the protein component, directly determined according to genomic and expression differences [23]. Nucleotides also comprised flavor substances, with certain nucleotide components (5′–GMP) presenting stronger flavor characteristics than MSG [24,25]. The interaction between nucleotides and MSG can greatly enhance the flavor of each, producing a much stronger flavor than that of the nucleotides and MSG alone or together [26,27]. The ten varieties of peanuts showed relatively high nucleotide contents (see Table 5), but with very different nucleotide components, similar to previous reports [28,29]. The electronic tongue response values revealed that the greatest taste difference among the ten peanut varieties was related to sourness, whereby Taihua 0311 had a low sour taste and Taihua 4 had a highly sour taste.

PCA is a statistical process that uses orthogonal transformations to convert the observed values of a group of potentially related variables into the values of a group of linearly unrelated variables called the principal components [30]. The deviation distribution of PCA1 of Taihua 4 and Taihua 9922 implied their taste specificities. The low contents of amino acids and 5′-nucleotides in Taihua 4 may lead to its flavor specificity, while the high contents of amino acids and 5′-nucleotides in Taihua 9922 may also influence its special taste characteristics. The loading plot of the PCA also confirmed that sourness made an important contribution to the flavor of peanuts. Sourness is mostly attributed to H^+^, and bitter materials can neutralize acids; therefore, the lowest bitter amino acid content in Taihua 4 may contribute to the highest sourness response value obtained via the electronic tongue for it.

The volatile components in the ten varieties ranged from 3.15% to 7.46%, with 9 to 18 kinds. Only 1-nonanol, nonanal and 3-methyl-undecane were detected in all ten peanut varieties. These results indicated great differences in the content and composition of volatile substances between the ten varieties of peanuts, and these results are similar to those previously reported [31]. The electronic nose response value revealed that the broad methane and broad alcohol were the main smell substances, and the smell of the peanuts was similar to that of the mixture of methane, carbon monoxide and nitrogen dioxide. The smell differences between the ten varieties of peanuts were mostly attributed to methane and sulfur organic substances. The PCA indicated that Taihua 0605-2 exhibited a particular odor characteristic based on the electronic nose signal, because its large deviation was distributed at PCA1, likely caused by its unique presence of 2,6,7-trimethyl-decane and high content of fats. More research on this special smell characteristic of Taihua 0605-2 should be carried out on variously processed and cooked peanuts.

## 5. Conclusions

The fat and protein contents of the ten peanut varieties were significantly (*p* < 0.05) different and negatively correlated. The amino acid contents of the peanuts were 20.08 g/100 g–27.18 g/100 g, and the 5′-nucleotide contents were 0.08 mg/g–0.14 mg/g. Forty-two kinds of volatile flavor compounds were detected. The electronic tongue and nose indicated that the dominant taste of peanuts was attributed to the mixture of sourness, astringency and bitterness, and the taste differences between the ten varieties of peanuts were due to sourness, with Taihua 4 and Taihua 9922 having special taste characteristics. The dominant smell of the peanuts was due to the mixture of methane, carbon monoxide and nitrogen dioxide, and the smell differences between the ten varieties of peanuts depended on methane and sulfur organic substances. Taihua 0605-2 exhibited special and strong smell characteristics. These results provide a theoretical basis for flavor research on peanuts, and guidelines for the utilization (fresh, processed or cooked) of these varieties.

## Figures and Tables

**Figure 1 foods-12-04380-f001:**
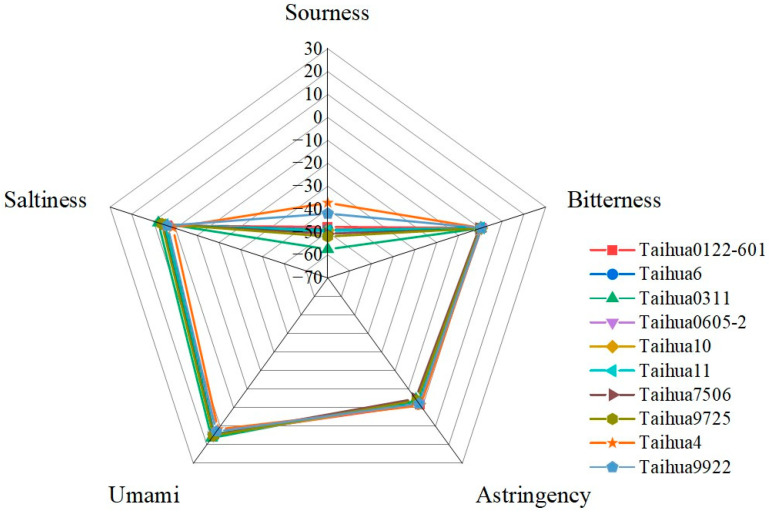
Radar map of the electronic tongue results for the ten peanut varieties.

**Figure 2 foods-12-04380-f002:**
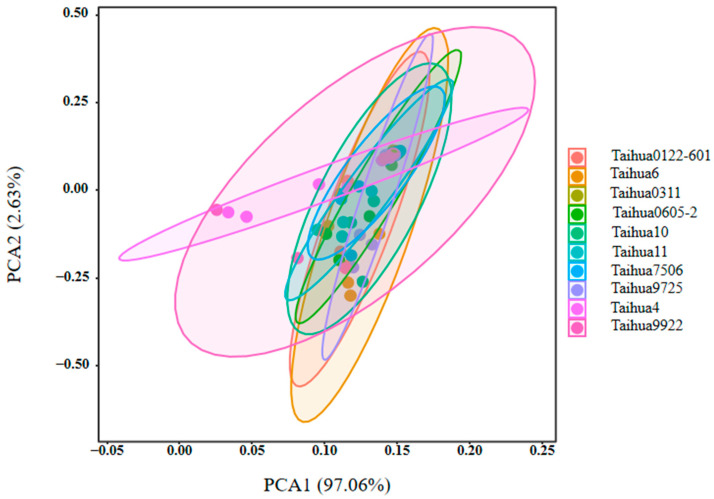
PCA of the electronic tongue results for the ten peanut varieties.

**Figure 3 foods-12-04380-f003:**
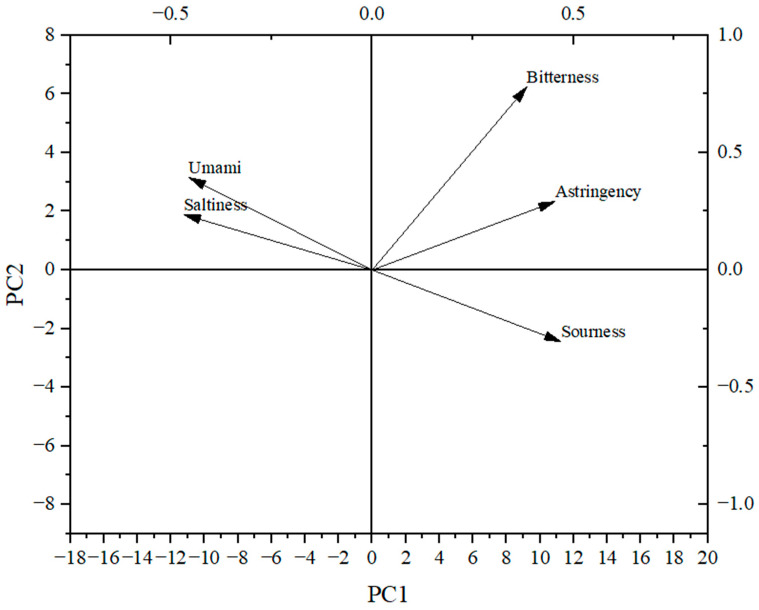
PCA loading plot of the electronic tongue results for the ten peanut varieties.

**Figure 4 foods-12-04380-f004:**
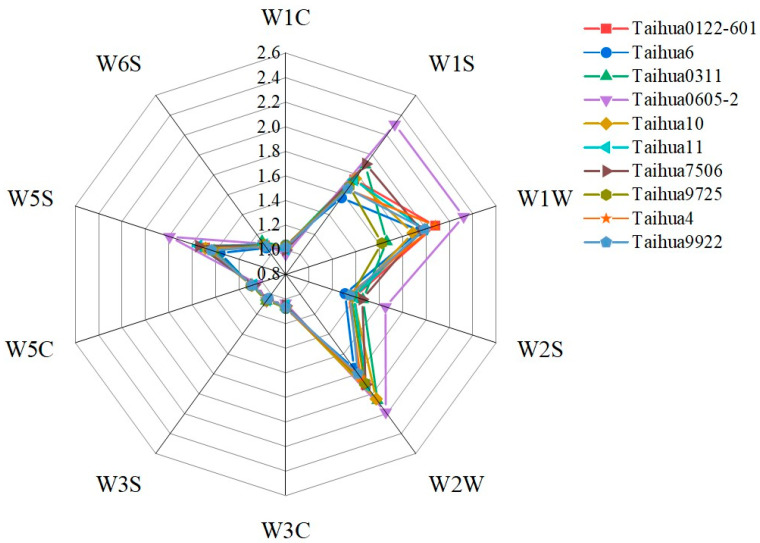
Radar map of the electronic nose results for the ten peanut varieties.

**Figure 5 foods-12-04380-f005:**
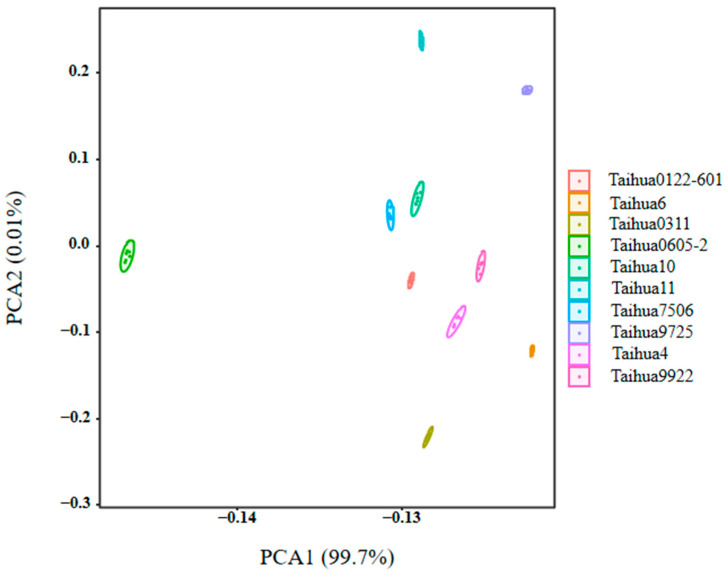
PCA of the electronic nose results for the ten peanut varieties.

**Figure 6 foods-12-04380-f006:**
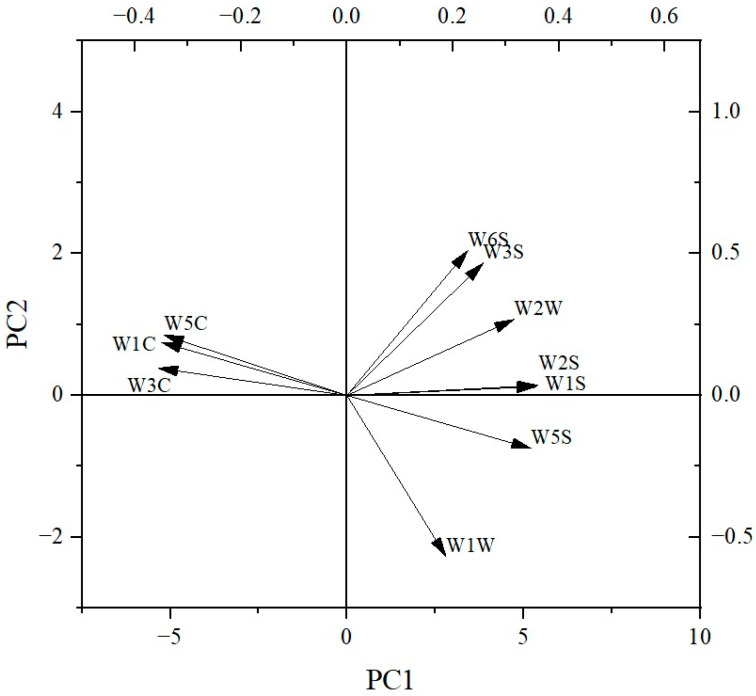
PCA loading plot of the electronic nose results for the ten peanut varieties.

**Table 1 foods-12-04380-t001:** Comparison of the water, fat and protein contents of the ten peanut varieties.

Relative Amount/%	Water Content	Fat Content	Protein Content
Taihua0122-601	5.152 ± 0.24 ^ab^	50.87 ± 1.67 ^abc^	27.54 ± 1.13 ^abc^
Taihua6	5.386 ± 0.23 ^ab^	48.24 ± 1.84 ^c^	29.90 ± 1.46 ^a^
Taihua0311	5.021 ± 0.16 ^ab^	50.07 ± 1.13 ^abc^	28.31 ± 1.28 ^ab^
Taihua0605-2	4.752 ± 0.14 ^b^	53.83 ± 1.53 ^a^	28.58 ± 1.42 ^a^
Taihua10	5.212 ± 0.27 ^ab^	51.02 ± 1.45 ^abc^	27.23 ± 1.18 ^abc^
Taihua11	5.483 ± 0.35 ^a^	48.47 ± 1.67 ^bc^	28.67 ± 1.87 ^a^
Taihua7506	5.274 ± 0.49 ^ab^	53.59 ± 2.15 ^a^	23.95 ± 1.45 ^c^
Taihua9725	5.129 ± 0.25 ^ab^	49.27 ± 1.57 ^bc^	27.96 ± 1.26 ^ab^
Taihua4	5.069 ± 0.18 ^ab^	52.38 ± 2.05 ^ab^	24.86 ± 1.06 ^bc^
Taihua9922	5.269 ± 0.29 ^ab^	50.43 ± 1.97 ^abc^	28.71 ± 1.76 ^a^

Different subscript letters in the same row for the same item indicate significant differences (*p* < 0.05).

**Table 2 foods-12-04380-t002:** Comparison of the free amino acid contents of the ten peanut varieties.

Relative Amount/g/100 g	Taihua0122-601	Taihua6	Taihua 0311	Taihua 0605-2	Taihua 10	Taihua 11	Taihua 7506	Taihua 9725	Taihua 4	Taihua 9922	Average
L-Aspartic	2.89 ± 0.31 ^d^	4.14 ± 1.23 ^a^	2.77 ± 0.33 ^d^	2.81 ± 0.21 ^d^	4.03 ± 1.18 ^ab^	4.06 ± 1.97 ^a^	3.55 ± 0.13 ^c^	3.64 ± 0.16 ^c^	3.71 ± 1.21 ^bc^	3.42 ± 0.61 ^c^	3.50
L-Glutamic	3.01 ± 0.98 ^bc^	3.24 ± 0.76 ^ab^	2.29 ± 0.88 ^d^	1.96 ± 0.23 ^d^	3.58 ± 1.22 ^a^	3.22 ± 0.86 ^b^	2.98 ± 0.98 ^bc^	3.02 ± 0.45 ^bc^	2.69 ± 0.11 ^c^	3.11 ± 0.24 ^b^	2.91
L-Serine	1.32 ± 0.87 ^de^	1.95 ± 0.92 ^ab^	1.15 ± 0.78 ^e^	1.56 ± 0.31 ^cd^	2.19 ± 0.56 ^a^	1.84 ± 0.99 ^bc^	1.23 ± 0.20 ^de^	1.39 ± 0.35 ^de^	1.41 ± 0.02 ^de^	1.51 ± 0.11 ^cd^	1.56
Glycine	1.22 ± 0.23 ^cd^	1.95 ± 0.14 ^a^	0.98 ± 0.02 ^d^	1.71 ± 0.38 ^ab^	1.86 ± 0.92 ^ab^	1.94 ± 0.19 ^a^	1.56 ± 0.29 ^bc^	1.82 ± 0.12 ^ab^	1.64 ± 0.18 ^ab^	1.02 ± 0.09 ^d^	1.57
L-Histidine	0.75 ± 0.09 ^abc^	0.98 ± 0.23 ^a^	0.64 ± 0.43 ^abcd^	0.87 ± 0.05 ^ab^	0.95 ± 0.19 ^a^	0.89 ± 0.44 ^ab^	0.48 ± 0.14 ^bd^	0.29 ± 0.11 ^d^	0.59 ± 0.32 ^bcd^	0.86 ± 0.13 ^ab^	0.73
L-Arginine	3.17 ± 1.43 ^bc^	3.68 ± 0.93 ^a^	3.09 ± 0.76 ^cd^	3.29 ± 1.32 ^bcd^	3.56 ± 1.22 ^ab^	3.50 ± 0.23 ^abc^	3.40 ± 0.76 ^abcd^	2.78 ± 0.78 ^de^	2.52 ± 0.98 ^e^	2.67 ± 0.68 ^e^	3.17
L-Threonine	0.31 ± 0.09 ^c^	0.67 ± 0.23 ^abc^	0.56 ± 0.45 ^abc^	0.49 ± 0.29 ^bc^	0.73 ± 0.45 ^ab^	0.91 ± 0.13 ^a^	0.64 ± 0.95 ^abc^	0.77 ± 0.55 ^ab^	0.82 ± 0.44 ^ab^	0.65 ± 0.08 ^abc^	0.66
L-Alanine	1.06 ± 0.34 ^cd^	1.49 ± 0.13 ^b^	0.78 ± 0.09 ^de^	1.86 ± 0.17 ^a^	1.08 ± 0.82 ^cd^	1.33 ± 0.91 ^bc^	0.88 ± 0.45 ^de^	0.67 ± 0.29 ^e^	1.27 ± 0.23 ^bc^	0.75 ± 0.14 ^de^	1.12
L-Proline	1.31 ± 0.93 ^a^	1.23 ± 0.14 ^ab^	0.88 ± 0.72 ^bcd^	0.77 ± 0.24 ^d^	1.14 ± 0.98 ^abc^	1.01 ± 0.04 ^abcd^	0.86 ± 0.12 ^cd^	0.97 ± 0.78 ^bcd^	0.76 ± 0.24 ^d^	0.98 ± 0.5 ^bcd^	0.99
L-Tyrosine	0.3 ± 0.09 ^abc^	0.21 ± 0.12 ^bc^	0.41 ± 0.23 ^abc^	0.58 ± 0.08 ^a^	0.59 ± 0.13 ^a^	0.09 ± 0.03 ^c^	0.28 ± 0.18 ^abc^	0.31 ± 0.08 ^abc^	0.54 ± 0.13 ^ab^	0.56 ± 0.23 ^a^	0.39
L-Valine	0.69 ± 0.34 ^cd^	0.63 ± 0.21 ^cd^	1.87 ± 0.37 ^a^	0.84 ± 0.34 ^cd^	0.97 ± 0.24 ^c^	0.62 ± 0.23 ^d^	1.54 ± 0.98 ^b^	0.60 ± 0.56 ^d^	0.61 ± 0.45 ^d^	0.71 ± 0.09 ^cd^	0.91
L-Methionine	0.11 ± 0.09 ^cd^	0.44 ± 0.39 ^abc^	0.28 ± 0.13 ^bcd^	0.56 ± 0.23 ^ab^	0.38 ± 0.16 ^bcd^	0.72 ± 0.56 ^a^	0.33 ± 0.23 ^bcd^	0.38 ± 0.14 ^bcd^	0.09 ± 0.06 ^d^	0.21 ± 0.11 ^cd^	0.35
L-Isoleucine	0.51 ± 0.34 ^bc^	0.93 ± 0.45 ^a^	0.82 ± 0.24 ^ab^	0.65 ± 0.45 ^ab^	0.55 ± 0.03 ^bc^	0.71 ± 0.67 ^ab^	0.86 ± 0.24 ^ab^	0.61 ± 0.45 ^ab^	0.21 ± 0.13 ^c^	0.76 ± 0.34 ^ab^	0.66
L-Leucine	0.67 ± 0.45 ^cd^	1.89 ± 0.23 ^a^	1.17 ± 0.56 ^b^	1.04 ± 0.35 ^b^	1.68 ± 0.45 ^a^	1.57 ± 0.19 ^a^	0.58 ± 0.34 ^d^	0.98 ± 0.18 ^bc^	0.85 ± 0.09 ^bcd^	1.06 ± 0.45 ^b^	1.15
L-Phenylalanine	1.37 ± 0.78 ^ef^	1.86 ± 0.45 ^bcd^	1.78 ± 0.12 ^bcd^	1.98 ± 0.34 ^bc^	2.10 ± 0.67 ^b^	1.57 ± 0.36 ^de^	1.64 ± 0.13 ^cde^	1.80 ± 0.08 ^bcd^	1.03 ± 0.15 ^f^	2.54 ± 0.36 ^a^	1.77
L-Lysine	1.89 ± 0.98 ^a^	1.15 ± 0.74 ^b^	1.23 ± 0.24 ^b^	0.97 ± 0.54 ^bc^	0.34 ± 0.06 ^e^	1.06 ± 0.35 ^b^	0.59 ± 0.21 ^de^	0.71 ± 0.45 ^cd^	0.43 ± 0.03 ^de^	0.64 ± 0.24 ^cde^	0.90
L-Tryptophan	0.88 ± 0.13 ^b^	0.74 ± 0.21 ^b^	0.82 ± 0.47 ^b^	0.78 ± 0.09 ^b^	0.71 ± 0.35 ^b^	1.68 ± 0.56 ^a^	0.29 ± 0.11 ^c^	0.58 ± 0.23 ^bc^	0.91 ± 0.43 ^b^	0.59 ± 0.17 ^bc^	0.80
Bitter	7.27 ± 3.52 ^bc^	10.41 ± 2.89 ^a^	9.65 ± 2.61 ^ab^	9.23 ± 3.08 ^ab^	10.19 ± 2.96 ^ab^	9.58 ± 2.68 ^ab^	8.83 ± 2.82 ^abc^	7.44 ± 2.30 ^abc^	5.9 ± 2.18 ^c^	8.81 ± 2.16 ^abc^	
Sweet	3.91 ± 1.53 ^b^	6.06 ± 1.42 ^a^	3.47 ± 1.34 ^b^	5.62 ± 1.15 ^ab^	5.86 ± 2.75 ^ab^	6.02 ± 2.22 ^a^	4.31 ± 1.89 ^ab^	4.65 ± 1.31 ^ab^	5.14 ± 0.87 ^ab^	3.93 ± 0.42 ^b^	
MSG-like	5.9 ± 1.29 ^ab^	7.38 ± 1.99 ^a^	5.06 ± 1.21 ^ab^	4.77 ± 0.44 ^b^	7.61 ± 2.40 ^a^	7.28 ± 2.83 ^a^	6.53 ± 1.11 ^ab^	6.66 ± 0.61 ^ab^	6.40 ± 1.32 ^ab^	6.53 ± 0.85 ^ab^	
Total	21.46 ± 4.23 ^b^	27.18 ± 7.25 ^a^	21.52 ± 5.45 ^b^	22.72 ± 6.23 ^b^	26.44 ± 7.44 ^a^	26.72 ± 6.34 ^a^	21.69 ± 5.25 ^b^	21.32 ± 7.26 ^b^	20.08 ± 5.48 ^b^	22.04 ± 5.65 ^b^	

Different subscript letters in the same row for the same item indicate significant differences (*p* < 0.05).

**Table 3 foods-12-04380-t003:** Comparison of the 5′-nucleotide contents of the ten peanut varieties.

5′-Nucleotides/mg/g	Taihua0122-601	Taihua6	Taihua0311	Taihua0605-2	Taihua10	Taihua11	Taihua7506	Taihua9725	Taihua4	Taihua9922
5-CMP	0.06 ± 0.03 ^a^	0.06 ± 0.03 ^a^	0.06 ± 0.04 ^a^	0.06 ± 0.0.1 ^a^	0.04 ± 0.04 ^c^	0.05 ± 0.01 ^b^	0.06 ± 0.07 ^a^	0.06 ± 0.01 ^a^	0.06 ± 0.03 ^a^	0.06 ± 0.06 ^a^
5′-UMP	ND	ND	0.05 ± 0.01 ^a^	ND	0.02 ± 0.01 ^b^	ND	ND	ND	ND	ND
5′-GMP	0.05 ± 0.03 ^a^	ND	ND	0.05 ± 0.01 ^a^	ND	0.05 ± 0.04 ^a^	0.04 ± 0.01 ^b^	ND	ND	0.05 ± 0.03 ^a^
5′-IMP	ND	ND	ND	ND	ND	ND	ND	ND	ND	ND
5′-AMP	0.03 ± 0.03 ^a^	0.02 ± 0.01 ^b^	0.02 ± 0.01 ^b^	0.02 ± 0.01 ^b^	0.02 ± 0.00 ^b^	0.02 ± 0.01 ^b^	ND	0.02 ± 0.01 ^b^	0.03 ± 0.01 ^a^	0.02 ± 0.01 ^b^
Total	0.14	0.08	0.13	0.13	0.08	0.12	0.10	0.08	0.09	0.13

Different subscript letters in the same row for the same item indicate significant differences (*p* < 0.05). ND indicates not detected.

**Table 4 foods-12-04380-t004:** Volatile compound contents of the ten peanut varieties.

Relative Amount/%	Taihua0122-601	Taihua6	Taihua0311	Taihua0605-2	Taihua10	Taihua11	Taihua7506	Taihua9725	Taihua4	Taihua9922
2-Octen-1-ol	ND	ND	ND	0.10 ± 0.04 ^bc^	0.32 ± 0.11 ^a^	ND	0.15 ± 0.07 ^b^	0.33 ± 0.08 ^a^	ND	ND
(E)-2-Octen-1-ol	0.10 ± 0.08 ^bc^	0.24 ± 0.06 ^b^	ND	ND	ND	ND	ND	ND	0.55 ± 0.28 ^a^	ND
(Z)-2-Octen-1-ol	ND	ND	ND	ND	ND	ND	ND	ND	ND	0.08 ± 0.08 ^a^
1-Nonanol	0.20 ± 0.08 ^cd^	0.22 ± 0.04 ^cd^	0.31 ± 0.13 ^cd^	0.18 ± 0.03 ^d^	0.62 ± 0.11 ^ab^	0.45 ± 0.06 ^abc^	0.26 ± 0.14 ^cd^	0.41 ± 0.18 ^bcd^	0.68 ± 0.06 ^a^	0.34 ± 0.08 ^cd^
2-Butyl-1-Octanol	ND	ND	ND	ND	ND	ND	ND	0.04 ± 0.04 ^a^	ND	ND
Nonanal	0.67 ± 0.40 ^bc^	1.28 ± 0.33 ^bc^	0.77 ± 0.10 ^bc^	0.37 ± 0.14 ^c^	2.72 ± 1.00 ^a^	1.59 ± 0.42 ^b^	0.92 ± 0.17 ^bc^	1.31 ± 0.44 ^bc^	1.41 ± 0.28 ^bc^	0.61 ± 0.11 ^bc^
2-Nonenal	0.08 ± 0.05 ^a^	ND	ND	ND	ND	ND	ND	ND	ND	0.06 ± 0.04 ^a^
(E)-2-Nonenal	ND	ND	0.08 ± 0.03 ^a^	ND	ND	ND	ND	ND	ND	ND
(Z)-2-Nonenal	ND	0.09 ± 0.07 ^a^	ND	ND	0.11 ± 0.03 ^a^	0.13 ± 0.07 ^a^	ND	0.11 ± 0.04 ^a^	0.10 ± 0.01 ^a^	ND
Decanal	0.09 ± 0.07 ^bc^	0.08 ± 0.04 ^bc^	0.15 ± 0.07 ^bc^	0.06 ± 0.06 ^bc^	0.31 ± 0.20 ^ab^	ND	ND	0.45 ± 0.24 ^a^	0.13 ± 0.04 ^bc^	ND
(E)-2-Decenal	ND	ND	ND	ND	ND	ND	0.10 ± 0.01 ^a^	ND	0.14 ± 0.08 ^a^	0.09 ± 0.08 ^ab^
(Z)-2-Decenal	0.09 ± 0.04 ^a^	ND	0.08 ± 0.08 ^a^	ND	ND	ND	ND	ND	ND	ND
2,4-Decadienal	ND	ND	ND	ND	ND	0.39 ± 0.27 ^a^	0.16 ± 0.08 ^ab^	ND	0.11 ± 0.06 ^b^	ND
(E,E)-2,4-Decadienal	ND	ND	0.08 ± 0.06 ^a^	ND	ND	ND	ND	ND	ND	ND
2-Undecenal	ND	ND	0.05 ± 0.02 ^a^	ND	ND	ND	ND	ND	ND	ND
2,4-Dimethyl-Benzaldehyde	ND	ND	ND	0.09 ± 0.04 ^ab^	0.14 ± 0.08 ^a^	ND	ND	ND	ND	0.15 ± 0.08 ^a^
Undecanal	ND	ND	ND	ND	0.03 ± 0.02 ^a^	ND	ND	ND	ND	ND
Formic Acid, Hexyl Ester	1.15 ± 0.92 ^bc^	1.52 ± 0.74 ^abc^	1.40 ± 0.24 ^bc^	2.04 ± 1.47 ^abc^	ND	3.10 ± 0.85 ^ab^	3.63 ± 0.91 ^a^	ND	2.01 ± 0.26 ^abc^	1.98 ± 1.33 ^abc^
Formic Acid, Heptyl Ester	0.08 ± 0.04 ^a^	ND	ND	ND	ND	ND	ND	ND	ND	ND
Formic Acid, Octyl Ester	ND	0.35 ± 0.34 ^a^	ND	ND	ND	0.37 ± 0.18 ^a^	0.21 ± 0.04 ^ab^	ND	ND	ND
Octanoic Acid, Methyl Ester	ND	0.15 ± 0.07 ^c^	ND	0.40 ± 0.20 ^bc^	0.68 ± 0.45 ^ab^	0.30 ± 0.16 ^bc^	ND	ND	0.89 ± 0.31 ^a^	ND
Nonanoic Acid, Methyl Ester	ND	ND	ND	0.04 ± 0.03 ^ab^	0.08 ± 0.01 ^a^	ND	ND	ND	0.10 ± 0.07 ^a^	ND
2-Methyl-Decane	0.11 ± 0.06 ^a^	0.08 ± 0.03 ^ab^	0.09 ± 0.08 ^a^	0.04 ± 0.03 ^ab^	ND	ND	ND	ND	ND	ND
3-Methyl-Decane	0.19 ± 0.08 ^a^	0.14 ± 0.07 ^a^	ND	0.17 ± 0.03 ^a^	ND	ND	ND	ND	ND	ND
4-Methyl-Decane	0.18 ± 0.16 ^a^	0.09 ± 0.06 ^ab^	ND	ND	ND	ND	ND	ND	ND	ND
5-Methyl-Decane	ND	ND	0.12 ± 0.06 ^a^	ND	ND	ND	ND	ND	ND	ND
2-Methyl-Undecane	ND	ND	ND	ND	ND	ND	ND	ND	0.18 ± 0.06 ^a^	ND
3-Methyl-Undecane	0.12 ± 0.11 ^ab^	0.05 ± 0.03 ^b^	0.08 ± 0.06 ^ab^	0.11 ± 0.06 ^ab^	0.25 ± 0.20 ^ab^	0.13 ± 0.04 ^ab^	0.06 ± 0.04 ^b^	0.10 ± 0.10 ^ab^	0.31 ± 0.18 ^a^	0.12 ± 0.07 ^ab^
4-Methyl-Undecane	ND	ND	ND	ND	ND	ND	ND	ND	0.10 ± 0.04 ^a^	ND
2,6-Dimethyl-Undecane	ND	0.19 ± 0.11 ^ab^	ND	ND	ND	ND	0.19 ± 0.13 ^ab^	0.38 ± 0.25 ^a^	ND	ND
2,6,7-Trimethyl-Decane	ND	ND	ND	0.07 ± 0.03 ^a^	ND	ND	ND	ND	ND	ND
2,6,8-Trimethyl-Decane	ND	0.11 ± 0.04 ^a^	ND	ND	ND	ND	ND	ND	ND	ND
2,6,11-Trimethyl-Dodecane	ND	ND	0.04 ± 0.03 ^a^	ND	ND	ND	ND	ND	ND	ND
3-Methyl-Tridecane	0.10 ± 0.07 ^ab^	0.16 ± 0.06 ^a^	0.09 ± 0.01 ^ab^	0.05 ± 0.03 ^ab^	0.11 ± 0.08 ^ab^	0.08 ± 0.07 ^ab^	ND	ND	0.14 ± 0.06 ^a^	0.04 ± 0.03 ^ab^
1-Dodecene	0.07 ± 0.04 ^ab^	ND	ND	ND	ND	ND	ND	ND	0.10 ± 0.06 ^a^	ND
(E)-2-Dodecene	ND	0.02 ± 0.01 ^a^	ND	ND	ND	ND	ND	ND	ND	ND
1-Tridecene	ND	ND	ND	ND	ND	ND	ND	ND	0.03 ± 0.02 ^a^	ND
6-Tridecene	ND	0.01 ± 0.01 ^a^	ND	ND	ND	ND	ND	ND	ND	ND
(R)-1-Methyl-5-(1-Methylethenyl)-Cyclohexene	ND	ND	ND	2.52 ± 1.19 ^a^	ND	ND	ND	0.02 ± 0.01 ^b^	ND	ND
o-Cymene	ND	ND	0.03 ± 0.01 ^c^	0.15 ± 0.09 ^a^	ND	ND	0.12 ± 0.08 ^b^	ND	ND	ND
Nonanoic Acid	ND	0.16 ± 0.11 ^ab^	ND	ND	ND	ND	ND	ND	0.26 ± 0.19 ^a^	ND
2-Pentyl-Furan	ND	ND	ND	ND	ND	ND	ND	ND	0.33 ± 0.12 ^a^	ND

Different subscript letters in the same row for the same item indicate significant differences (*p* < 0.05). ND indicates not detected.

**Table 5 foods-12-04380-t005:** Volatile compounds responding to the 10 sensors of the electronic nose.

Sensor No.	Sensing Species	Reference Substance
**W1C**	aromatic	toluene
**W1S**	broad methane	methane
**W1W**	sulfur organic	hydrogen sulfide
**W2S**	broad alcohol	carbon monoxide
**W2W**	sulf–chlor	hydrogen sulfide
**W3C**	aromatic	benzene
**W3S**	methane–aliph	methane
**W5C**	arom–aliph	propane
**W5S**	broad-range	nitrogen dioxide
**W6S**	hydrogen	hydrogen

## Data Availability

The data used to support the findings of this study can be made available by the corresponding author upon request.

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
