# Peer review of "Flavor Characteristics of Ten Peanut Varieties from China"

_foods, 2023, doi:10.3390/foods12244380_

Round 1

Reviewer 1 Report

Comments and Suggestions for Authors

This study determines flavor and taste of various 10 species of peanuts by using e-nose and e-tongue. The topic is interesting, and the experimental design and techniques used are reliable in achieving the study's objectives. However, I concern that the English in this manuscript, especially in some sentences, should be rewritten or edited by a native speaker to enhance clarity and avoid potential misunderstandings. Additionally, there are some issues that need to be addressed by the author. Here are the comments.

Major concerns

- The topic is too broad; it seems like this is a review. I suggest the author specify it more clearly.

- Determination of volatiles should be provided in the text. Also, the measurement/calculation should be re-checked as I mentioned in details below.

- For PCA, loading plots should be provided to gain more data and lead to more discussion.

- The discussion should be improved (not only report the data)., the findings should be scoped (not only concluded that it’s difference).

ABSTRACT

- Line14: Changed? Or contained? (original have), please check for accuracy.

- Line17: Please add (P < 0.05) or a significant sign for better identification. Not only this sentence, this applies should be done to all sentences indicating higher/lower values; if they are statistically significant, the symbol should be included.

- Line18: As per my experiences, I think the unit of amino acid content that appeared at % is not familiar. It should be g/100g or mg/100 g or etc. If it appeared at % >> % of what?

- Line21: The abbreviation likes CMP, AMP, etc. should come with the full name since it appeared at the first time.

- Line22-24: I suggested author to rewrite this sentence related to the volatile results since these sentences are confusing, lots of plain data. The author might change into the average/ which one is highest/ lowest or other highlighted results.

- Line27: This sentence is confusing. It means the sourness among all samples is difference? Or All samples provided the sourness at dominant? Please rewrite it.

- I suggested to end up the abstract with overall conclusion or further application/suggestions from the findings.

INTRODUCTION

- The introduction should gain more information related to the objectives of this work. Such as how’s important of flavor/taste of the peanuts (it linked to the consumer acceptance)? How’s the variation among each species and what’s the effect? The flavor/taste of other peanuts which appeared in previous studies should be described in brief. How’s the e-nose/e-togue provided more information related to traditional technique, etc.

- As author stated that the findings will be useful for further utilization >> How?

MATERIALS AND METHODS

- Please identify clearly sample lots/sets and replicates done in this MS to state that the sample size is suitable. All samples were obtained for the same pilot farm and assumable, may have a limited representation of the population. A sample with broader representation or specified constraints would have been necessary to achieve the objectives for this study.

- The sample codes appear to be confusing. I suggest the author to reorganize or assign them with simple numbering, such as 'Taihua1,' 'Taihua2,' 'Taihua3,' etc., unless they have specific meanings that I am not aware of.

- Line80: Please add the conditions or more details related to “dried in natural conditions” since it reflects to flavor/taste, which further determines.

- Line81: Please state how long the sample was collected until analyzed since it also reflect to the result as well.

- Line94: The procedures, for at least, the sample preparation, condition used, of e-nose and e-tongue should be provided more details.

- Important!!!! There was no “determination of volatiles” appeared in this part.

RESULTS AND DISCUSSION

- Table2: Why author determine the average values of each amino acids, but not determine for other parameters?

- Table3: Again as I mentioned above! There are a lot of abbreviations. Please provide the full name at the first time appearing.

- Table4: How’s the volatiles were calculated. It seems like it reported as relative amount (%), but why overall amount in each sample was very less (for example Taihua0 were only 2%)?????. Please clarify this point. Moreover, how’s to preparation the samples, measurement and calculations the volatiles should provided in the M&M part.

- I suggested author changes this part by combing “results and discussion”, then, can discuss parameters by parameters. Now, the volatiles did not discuss enough. It mixed with the e-tongue and e-nose results, which did not discuss in details yet.

- The PCA results of them should discuss clearly at another paragraph. Moreover. Author should provide the loading plots of them to identified the parameters used. This can gain more information (because having only score plots gives only how much difference among samples but do not how’s it differ)

- For the discussion, how’s different among tastes/flavors as indicated by e-tongue, e-nose, volatiles and non-volatiles should be discussed (this will be achieved by having the loading plots).

CONCLUSION

- The conclusion should be more specific/precise. Author can improve after revise the discussion as per I suggested. (This included abstract as well).

Author Response

Response to comments 1

Dear reviewer,

Thank you for your valuable comments and suggestions, it is very professional and means a lot to us. We improved the quality of this paper followed you opinion, and changed the details point by point. Here we response your comments as below.

This study determines flavor and taste of various 10 species of peanuts by using e-nose and e-tongue. The topic is interesting, and the experimental design and techniques used are reliable in achieving the study's objectives. However, I concern that the English in this manuscript, especially in some sentences, should be rewritten or edited by a native speaker to enhance clarity and avoid potential misunderstandings. Additionally, there are some issues that need to be addressed by the author. Here are the comments.

Answer: thank you for your comment, we already rewritten the paper and checked the language again.

Major concerns

- The topic is too broad; it seems like this is a review. I suggest the author specify it more clearly.

Yes, we have changed the title of the paper and made it more specific.

- Determination of volatiles should be provided in the text. Also, the measurement/calculation should be re-checked as I mentioned in details below.

Yes, we have added the main part of methods to the paper.

- For PCA, loading plots should be provided to gain more data and lead to more discussion.

Yes, we have added the loading plots of PCA, and leaded to more discussion.

- The discussion should be improved (not only report the data)., the findings should be scoped (not only concluded that it’s difference).

Yes, we have improved the discussion and made it more specific.

ABSTRACT

- Line14: Changed? Or contained? (original have), please check for accuracy.

Yes, it is changed. It means the great different of fat and protein contents between ten varieties.

- Line17: Please add (P < 0.05) or a significant sign for better identification. Not only this sentence, this applies should be done to all sentences indicating higher/lower values; if they are statistically significant, the symbol should be included.

Yes, we already added the sign through whole paper.

- Line18: As per my experiences, I think the unit of amino acid content that appeared at % is not familiar. It should be g/100g or mg/100 g or etc. If it appeared at % >> % of what?

Yes, we already changed the unit through whole paper.

- Line21: The abbreviation likes CMP, AMP, etc. should come with the full name since it appeared at the first time.

Yes, we already added the full name.

- Line22-24: I suggested author to rewrite this sentence related to the volatile results since these sentences are confusing, lots of plain data. The author might change into the average/ which one is highest/ lowest or other highlighted results.

Yes, we already rewritten this sentence.

- Line27: This sentence is confusing. It means the sourness among all samples is difference? Or All samples provided the sourness at dominant? Please rewrite it.

Yes, we already cleared the confusion.

- I suggested to end up the abstract with overall conclusion or further application/suggestions from the findings.

Yes, we already added the information you suggested.

INTRODUCTION

- The introduction should gain more information related to the objectives of this work. Such as how’s important of flavor/taste of the peanuts (it linked to the consumer acceptance)? How’s the variation among each species and what’s the effect? The flavor/taste of other peanuts which appeared in previous studies should be described in brief. How’s the e-nose/e-togue provided more information related to traditional technique, etc.

Yes, we added the information that you suggested.

- As author stated that the findings will be useful for further utilization >> How?

Yes, we added the information that you suggested.

MATERIALS AND METHODS

- Please identify clearly sample lots/sets and replicates done in this MS to state that the sample size is suitable. All samples were obtained for the same pilot farm and assumable, may have a limited representation of the population. A sample with broader representation or specified constraints would have been necessary to achieve the objectives for this study.

Yes, we already added the information.

- The sample codes appear to be confusing. I suggest the author to reorganize or assign them with simple numbering, such as 'Taihua1,' 'Taihua2,' 'Taihua3,' etc., unless they have specific meanings that I am not aware of.

Thank you for your suggestion, the name of varieties was already existed before our comparison, we also suggest to keep the name for readability and public acceptance of paper.

- Line80: Please add the conditions or more details related to “dried in natural conditions” since it reflects to flavor/taste, which further determines.

Yes, we added the information.

- Line81: Please state how long the sample was collected until analyzed since it also reflect to the result as well.

Yes, we added the information.

- Line94: The procedures, for at least, the sample preparation, condition used, of e-nose and e-tongue should be provided more details.

Yes, we added the information.

- Important!!!! There was no “determination of volatiles” appeared in this part.

 Yes, we added the information.

RESULTS AND DISCUSSION

- Table2: Why author determine the average values of each amino acids, but not determine for other parameters?

Yes, actually we can determine more parameters, but the amino acids are very important even dominant flavor substance, the content and average content could reflect its flavor character, because every amino acid have its flavor.

- Table3: Again as I mentioned above! There are a lot of abbreviations. Please provide the full name at the first time appearing.

Yes, we already checked the whole paper, to make sure abbreviations have full name at the first time appearing.

- Table4: How’s the volatiles were calculated. It seems like it reported as relative amount (%), but why overall amount in each sample was very less (for example Taihua0 were only 2%)?????. Please clarify this point. Moreover, how’s to preparation the samples, measurement and calculations the volatiles should provided in the M&M part.

The calculation of volatiles from peanuts was according the peak area normalization method. We already added the information in the M&M part. The data were relatively small, maybe is the matter of fresh material, because the fresh material have far less volatile substance than processed ones.

- I suggested author changes this part by combing “results and discussion”, then, can discuss parameters by parameters. Now, the volatiles did not discuss enough. It mixed with the e-tongue and e-nose results, which did not discuss in details yet.

Thank you for your suggestion, we thought the merge of result and discussion seem a little disorder, but we enriched the discussion part.

- The PCA results of them should discuss clearly at another paragraph. Moreover. Author should provide the loading plots of them to identified the parameters used. This can gain more information (because having only score plots gives only how much difference among samples but do not how’s it differ)

Yes, we added the information that you suggested.

- For the discussion, how’s different among tastes/flavors as indicated by e-tongue, e-nose, volatiles and non-volatiles should be discussed (this will be achieved by having the loading plots).

 Yes, we added the information that you suggested.

CONCLUSION

- The conclusion should be more specific/precise. Author can improve after revise the discussion as per I suggested. (This included abstract as well).

Yes, we improved the conclusion more specific.

Thank you again for your suggestions, it’s very helpful.

Chunlu Qian

Yangzhou University

Reviewer 2 Report

Comments and Suggestions for Authors

The research article titled “Flavor character of peanuts as influenced by variety” discusses the physicochemical properties (moisture, protein and fat content) of peanuts, free amino acids and 5’-nucleotide content and taste and smell characteristics by e-nose and e-tongue. Although the research provides insight into ten local varieties of peanuts grown in the Jiangsu province of China, it lacks novelty, and the topic of this research is specific and limited, which would invite a limited readership. The grammatical accuracy in the entire manuscript needs to be thoroughly rechecked and improved for better understanding and clarity by readers.
Below are some other comments:
Abstract:
The availability of a variety of peanuts under this study on a global scale could be discussed if these varieties are locally available or globally produced.
The authors discussed the results of the study without elaborating on the nomenclature of the varieties considered for this study. E.g. L-17 “Taihua6 exhibited the highest protein content”; here, Taihua6 should have been previously defined.
Highlight the significance of Formic acid hexyl ester and 5’-nucleotide in peanuts in this study.
Introduction:
L37-39: Please discuss the quantity of exports or if consumed domestically and cite a reference for the same.
Materials and methods:
Section 2.1:
The authors have not provided details of the storage conditions of peanuts after procurement from the Jiangsu Academy of Agricultural Sciences.
L80: Drying conditions to be mentioned.
L81: Material and GSM of plastic bags should be mentioned.
Section 2.2:
L84: Dimensions and other specifications of the drying box are to be mentioned; if it was an equipment then kindly mention the make, model and other details.
Was the drying done until a constant weight was obtained or for 24 hours?
L86: Mention the amount of sample taken, sample preparation method and if it was taken after removal of moisture or along with moisture content? The authors in the literature cited does not employ or describe any Soxhlet extraction method in their study.
Section 2.3
To be re written for better understanding of the readers.
Table 2: L121: Table header to be revised
Table 2: What does average in this table signify?
Discussion:
L228–230: Needs more clarification as to what PCA1 denotes and how the results relate to the physical attributes.
Conclusion:
A future scope of study and industrial application could be included.

Author Response

Response to comments 2

Dear reviewer,

Thank you for your valuable comments and suggestions, it is very professional and means a lot to us. We improved the quality of this paper followed you opinion, and changed the details point by point. Here we response your comments as below.

The research article titled “Flavor character of peanuts as influenced by variety” discusses the physicochemical properties (moisture, protein and fat content) of peanuts, free amino acids and 5’-nucleotide content and taste and smell characteristics by e-nose and e-tongue. Although the research provides insight into ten local varieties of peanuts grown in the Jiangsu province of China, it lacks novelty, and the topic of this research is specific and limited, which would invite a limited readership. The grammatical accuracy in the entire manuscript needs to be thoroughly rechecked and improved for better understanding and clarity by readers.

Below are some other comments:  

Answer: thank you for your comment, we already improved the paper followed your suggestions.

Abstract:

The availability of a variety of peanuts under this study on a global scale could be discussed if these varieties are locally available or globally produced.

Yes, we already added the information that you suggested.

The authors discussed the results of the study without elaborating on the nomenclature of the varieties considered for this study. E.g. L-17 “Taihua6 exhibited the highest protein content”; here, Taihua6 should have been previously defined.

Yes, we already added the information that you suggested, the name of peanuts already existed before this study, and the peanuts already widely planted, this is first time for the comparison of their flavor.

Highlight the significance of Formic acid hexyl ester and 5’-nucleotide in peanuts in this study.

Yes, we already pay attention on formic acid hexyl ester and 5’-nucleotide, but their were not the main different substances, kind of stable.

Introduction:

L37-39: Please discuss the quantity of exports or if consumed domestically and cite a reference for the same.

Yes, we added the information you suggested.

Materials and methods:

Section 2.1:

The authors have not provided details of the storage conditions of peanuts after procurement from the Jiangsu Academy of Agricultural Sciences.

Yes, we added the information you suggested.

L80: Drying conditions to be mentioned.

Yes, we added the information you suggested.

L81: Material and GSM of plastic bags should be mentioned.

Yes, we added the information you suggested.

Section 2.2:

L84: Dimensions and other specifications of the drying box are to be mentioned; if it was an equipment then kindly mention the make, model and other details.

Yes, we added the information you suggested.

Was the drying done until a constant weight was obtained or for 24 hours?

Yes, we changed the statement, its till a constant weight obtained, it’s usual use 24h.

L86: Mention the amount of sample taken, sample preparation method and if it was taken after removal of moisture or along with moisture content? The authors in the literature cited does not employ or describe any Soxhlet extraction method in their study. 

Yes, we added the information you suggested.

Section 2.3

To be re written for better understanding of the readers.

Yes, we already rewrote this part.

Table 2: L121: Table header to be revised

Yes, we thought no mistake existed in table head of table 2.

Table 2: What does average in this table signify?

It’s the average content of amino acid in ten varieties of peanuts.

Discussion:

L228–230: Needs more clarification as to what PCA1 denotes and how the results relate to the physical attributes.

Yes, we added the information you suggested.

Conclusion:

A future scope of study and industrial application could be included.

Yes, we added the information that you suggested.

Thank you again for your suggestions, it’s very helpful.

Chunlu Qian

Yangzhou University

Reviewer 3 Report

Comments and Suggestions for Authors

The manuscript of the article includes the relevant Introduction chapter and the implementation of the experimental work. The analytical part (instrumental and statistical) corresponds to the performance of the experiment. The obtained results are presented in 5 tables and 4 figures (from radar plots to PCA analysis). A discussion of the numerous results obtained with a total of 28 references used and the conclusions are satisfactory. A few "cosmetic" corrections are noted in the attached manuscript review.

Author Response

Response to comments 3

Dear reviewer,

Thank you for your valuable comments and suggestions, it is very professional and means a lot to us. We improved the quality of this paper followed you opinion, and changed the details point by point. Here we response your comments as below.

The manuscript of the article includes the relevant Introduction chapter and the implementation of the experimental work. The analytical part (instrumental and statistical) corresponds to the performance of the experiment. The obtained results are presented in 5 tables and 4 figures (from radar plots to PCA analysis). A discussion of the numerous results obtained with a total of 28 references used and the conclusions are satisfactory. A few "cosmetic" corrections are noted in the attached manuscript review.

Yes, we improved the quality of paper according to you suggestion, please find details in the new version of paper.

Thank you again for your suggestions, it’s very professional and helpful.

Chunlu Qian

Yangzhou University

Round 2

Reviewer 1 Report

Comments and Suggestions for Authors

The MS has been revised as suggested before.

However, the quality of them is fair. In my opinion, I believe that certain parts of the manuscript, particularly the discussion section, need substantial revision. It is overly long, and sentences should be simplified. Additionally, the results should not be repeated excessively.

While the study's findings are adequately presented, there is a lack of interpretation and contextualization for the reader. More references are also needed to enrich the manuscript.

I've provided some suggestions, but I believe the entire paper should be rewritten to enhance its coherence and eliminate redundancy from the beginning.

Author Response

Dear reviewer,

Thank you for your review, the comments and suggestions provided were very professional and helpful for the improvement of the paper.

The MS has been revised as suggested before.

Yes, we carefully revised the paper according the suggestion from 3 reviewers.

However, the quality of them is fair. In my opinion, I believe that certain parts of the manuscript, particularly the discussion section, need substantial revision. It is overly long, and sentences should be simplified. Additionally, the results should not be repeated excessively.

Thank you for your suggestion, we already improve the quality as your comments, and simplified the discussion.

While the study's findings are adequately presented, there is a lack of interpretation and contextualization for the reader. More references are also needed to enrich the manuscript.

Yes, we already added references to make the paper more readable.

I've provided some suggestions, but I believe the entire paper should be rewritten to enhance its coherence and eliminate redundancy from the beginning.

Yes, we already rewrote the paper to enhance its coherence and eliminate redundancy. Some content of the paper were added according the suggestion from reviewer, so we presented the paper in a simpler way.

Thank you for your comments.

Chunlu Qian

Yangzhou University

Reviewer 2 Report

Comments and Suggestions for Authors

The authors have attended to the queries raised by the reviewer satisfactorily.

Author Response

Dear reviewer,

Thank you for your comments and agreement.

The authors have attended to the queries raised by the reviewer satisfactorily.

Best wishes.

Chunlu Qian

Yangzhou Universtiy